



# 1 InSAR monitoring of Arctic land fast sea ice deformation using
# 2 L-band ALOS-2, C-band Radarsat-2 and Sentinel-1

Zhaohua Chen[1], Benoit Montpetit[1], Sarah Banks[1], Lori White[1], Amir Behnamian[1], Jason Duffe[1],
Jon Pasher[1]
Science and Technology Branch, Environment and Climate Change Canada, Ottawa, Ontario, K1S 5B6, Canada
*Correspondence to*: Zhaohua Chen (zhaohua.chen@canada.ca)
**Abstract.** Arctic amplification is accelerating changes in sea ice regimes in the Canadian Arctic with later freeze-up
and earlier melt events, adversely affecting Arctic wildlife and communities that depend on the stability of the sea ice
conditions. To monitor both the rate and impact of such change, there is a need to accurately measure sea ice
deformation, an important component for understanding ice motion and polar climate. This paper presents
Interferometric Synthetic Aperture Radar (InSAR) monitoring of Arctic landfast sea ice deformation as a result of
thickness changes measured from ice draft and surface height using C-band Radarsat-2, Sentinel-1 and L-band ALOS-
2. The small baseline subset (SBAS) approach was explored to process time series observations for retrieval of
temporal deformation changes over the winter. Sea ice deformation (subsidence and uplift in the range of -32-57 cm)
detected from satellite SAR data in Cambridge Bay, Nunavut, Canada during the winter of 2018-2019 was found to
be in a range of values corresponding to the ice draft growth (30-62 cm) measured from an in-situ ice profiler. The
trends of InSAR observations from Sentinel-1 were also consistent with ice surface height changes along two ground
tracks detected from ICESat-2. SAR backscatter from Sentinel-1 also corresponded to the surface height with strong
correlation coefficient (0.49-0.83). High coherence over ice from C-band was maintained over a shorter acquisition
interval than L-band due to temporal decorrelation.

## 24 1 Introduction

Arctic ice has been thinning and retreating due to global warming in recent years (Tucker et al. 2001; Perovich et al.,
2008; Kwok et al., 2009; Wadhams, 2012). Given the vast sea ice extent and increasing commercial interest in the
Arctic, it is important to map sea ice changes (Dawson et al., 2020). SAR satellite imagery are used routinely to
identify and map sea ice changes due to its weather-independent capability (Zakhvatkina et al., 2019). Several C-band
SAR satellites are also used to provide operational products of sea ice types, extent and concentration due to the
availability of large amount of imagery (Arkett et al., 2006; Tivy, et al., 2011; Zakhvatkina et al., 2019). With
increasing data available from other SAR sensors, X- and L-band have also been found to provide complementary



information for sea ice identification (Drinkwater et al., 1991; Eriksson et al., 2010; Casey et al. 2016; Johansson et
al., 2018).
In addition to information on sea ice extent, types, and concentration, measurements of sea ice thickness and
kinematics are critical for understanding the impacts of the Arctic climate change (Zwally et al., 2008; Kwok, 2010).
Measurements of these parameters using satellite images are challenging because monitoring dynamic sea ice
processes requires high spatial and temporal resolution data (Kwok, 2010). A direct measurement of ice thickness can
be made through ice draft observations from in-situ sensors. Ice draft is a measurement of the ice thickness below the
waterline and may be used as a proxy for total ice thickness (Fisse et al., 2008). Currently, sea ice thickness is not
directly measured from space, but indirect measurements including surface height variations and freeboard have been
used to provide information related to thickness (Petty et al., 2016; Cafarella et al., 2019). Traditional methods like
ice core drilling and sonar or radar systems are used to obtain ice information below the water line, making them both
complex and resource intensive to implement over a large area. By comparison, satellite-based methods are cost-
effective and have demonstrated the potential for ice surface height measurements over large areas. Surface height
information can be measured by the radar altimeter from the Cryosat-2 satellite (Bouffard et al., 2018; Tilling et al.,
2018) or the lidar altimeter from Ice, Cloud, and land Elevation Satellite (ICESat-1 and ICESat-2) (Kwok et al., 2019).
Compared to the 369-day repeat cycle of Cryosat-2, ICESat-2 has a 91-day orbit cycle, providing higher temporal
frequency data for monitoring Arctic sea ice since its launch in 2018 (Kwok et al., 2019). ICESat-2 measurements
theoretically represent the vertical height of ice/snow surface above sea level. With close to 1-2 cm accuracy, surface
height measurements from laser pulses bouncing off the ice surface can be very important for studying ice conditions
in the Arctic (Kwok et al., 2016; Kwok et al., 2019). However, measurements from these altimeters are only point
data along the orbit, limiting their spatial coverage.
Surface height changes may also be measured indirectly by Interferometric Synthetic Aperture Radar (InSAR), a
technique that measures the line-of-sight (LOS) motion of a target based on the phase differences between two SAR
images. In fact, ice deformation has been investigated using this method near the Beaufort Sea coast (Meyer et al.,
2011; Dammann et al., 2016; Dammann et al., 2018a; Dammann et al., 2018b) and the Baltic sea coast (Dammert et
al., 1998; Berg et al., 2015; Marbouti et al., 2017). It is believed that most of the InSAR phase change over a coherent
area are the results of vertical or horizontal movements of the sea ice (Dammert et al., 1998; Meyer et al., 2011; Berg
et al., 2015; Dammann et al., 2018a). The deformation is an important component of ice motion. It is attributed to
thermal expansion and contraction or dynamic deformation (e.g., compression of the ice driven by winds, sea level
tilt, currents, or internal ice stress (Dammann et al., 2016; Marbouti et al., 2017). Aided with inverse modeling, InSAR
may be used to determine ice deformation modes (Dammann et al., 2016), rates (Meyer et al., 2011), dynamics (e.g.,
associated stress and fracture patterns) (Dammert et al., 1998; Berg et al., 2015; Marbouti et al., 2017; Dammann et
al., 2018b; Dammann et al., 2019), and topography (Dammann et al., 2018b; Dierking et al., 2017). However, it is
difficult to verify InSAR measurements of ice deformation due to the cost associated with the installation of equipment
for in-situ measurements and the difficulty of accessing ice away from the shore.



InSAR measurements of sea ice deformation can be affected by several factors including coherence, frequency, sea
ice thickness and others (e.g. wetness, salinity). Interferometric information is useful only if the phase differences are
from coherent areas. Coherence (value ranges from 0 to 1) is used to describe the quality of an interferogram which
contains phase changes. Higher coherence values indicate higher phase correlation and less noise in interferometric
phase. Usually, targets that remain unchanged during the time interval of two image acquisitions have high coherence.
On the other hand, low coherence values can be a result of temporal decorrelation, where there is little to no correlation
observed among groups of pixels observed at the same location, through time. This can be a result of either significant
motion occurring between acquisitions (measured relative to the radar wavelength) and/or due to a low temporal
frequency of acquisitions. Notably, temporal decorrelation can be significant in areas dominated by ice, especially
floating ice, due to relatively quick horizontal movements as a result of wind/currents/waves (Berg et al., 2015;
Dammann et al., 2016; Marbouti et al., 2017). Though generally speaking, for most natural surfaces, the longer the
time interval between acquisitions, the lower the coherence. Loss of coherence is also dependent on the wavelength.
Decorrelation is more pronounced at shorter wavelengths such as X-band (2.5-4 cm), and C- (4-8 cm) than longer
wavelengths such as L-band (15-30 cm). Many InSAR studies confirm improved coherence in L-band SAR data
(Rosen et al., 2000; Dammann et al., 2016; Meyer et al., 2011). Wavelength also affects InSAR results because it
impacts how incident microwaves interact with the snow/ice surface and ice volume (Kwok, 2010; Berg et al., 2015;
Dammann et al., 2016). Thus, the sensitivity of SAR to sea ice deformation is dependent on the penetration depth of
a respective wavelength (Dierking and Busche, 2006; Dierking and Dall, 2007; Johansson et al., 2018). Longer
wavelength microwaves can penetrate deeper than shorter wavelength microwaves (Johansson et al., 2018). InSAR
coherence is generally higher and more stable over stationary landfast ice than drifting ice (Dammann et al., 2016;
Dierking et al., 2017; Marbouti et al., 2017; Dammann et al., 2018a; Dammann et al., 2018b). As a result, current
efforts to monitor sea ice deformation using satellite data have therefore been mainly limited to landfast ice (Dammert
et al., 1998; Meyer et al., 2011; Berg et al., 2015). Landfast sea ice is an essential element of Arctic coastal sea ice
systems, and its stability is critical for transportation and activities using ice as a platform (Dammann et al., 2016).
Understanding the response of SAR signals to changes in surface height above the sea level and thickness changes
below the sea level can help determine the patterns and sources of deformation, a critical first step to better
investigating the impacts of climate change on sea ice. To calculate the ice deformation over a certain period with
time series SAR images, small baseline subset (SBAS) (Berardino et al., 2002; Usai, 2003) may be applied. The SBAS
method can connect interferograms spanning in different time intervals and generate cumulative measurements of
movements while limiting and correcting effects from temporal decorrelation , atmosphere and topography (Berardino
et al., 2002; Usai, 2003; Samsonov et al., 2014; Chen et al., 2016; Chen etal., 2020). Once the deformation is derived,
validation of InSAR results is important. The validation of InSAR results can help improve the generation of products
related to ice stability, thickness and ice conditions. With frequent SAR acquisitions, InSAR derived measurements
may provide the information with increased spatial and temporal coverage that is not available from the in-situ ice
profiler and ICESat-2. InSAR deformation is the projection of the three-dimensional ice movement along the LOS
direction, containing both horizontal and vertical information. Assuming that fast ice does not move with wind and
sea current, and the vertical deformation over stationary fast ice is the dominant movement due to changes of ice



thickness, InSAR LOS deformation over coherent fast ice may reflect surface height variations measured by ICESat-
2 as well as the ice draft measured from the in-situ ice profiler, which could be verified via correlation analyses. From
this perspective, we investigated InSAR derived deformation information observed from SAR satellites and compared
with surface height changes measured from satellite lidar altimeter and ice draft measured from the in-situ ice profiler
in Cambridge Bay, Nunavut, Canada. The surface deformation measurements from SAR sensors including L-band
ALOS-2, C-band Radarsat-2 and Sentinel-1 were generated using InSAR techniques. In this paper, we evaluated the
sensitivity of SAR backscatter to sea ice surface height changes, investigated changes and differences in coherence
between L- and C- band SAR data over arctic sea ice, and assessed how InSAR observations correlate to surface
height variations and ice draft growth.


**2 Study area and data**
Cambridge Bayis located on Victoria Island, Nunavut, Canada. It is directly connected to the Arctic Ocean and is
partially covered by ice for most of the year. This study monitored the sea ice covering the Northwest Passage through
the Arctic Ocean between Dease Strait (located on the left side of Cambridge Bay) and Queen Maud Gulf (located on
the right side of Cambridge Bay) (Figure 1). During winter months, the entire area is covered by landfast first year ice
(FYI) which varies in thickness depending on the time of year (Figure 2). Some multi-year ice (MYI) may flow from
Victoria Strait to the east into the Queen Maud Gulf but regional climate charts available from the Canadian Ice
Service Data Archive (CISDA) (Tivy et al., 2011) showed that the dominant ice type was FYI with varying surface
roughness and thickness in the study area (Figure 2).
For this area, ice draft information was provided by an upward looking sonar water ice profiler (SWIP), deployed by
Ocean Networks Canada. The ice profiler uses acoustics to measure the ice thickness, known as ice draft, and is
mounted on the ocean floor at a depth of 6 m near the Cambridge Bay dock (Ocean Networks Canada, 2019) (Figure
1).  A review of the observatory data of ice draft from previous years indicates that ice was formed on 15 October, 7
October, 1 October, and 8 October in 2016, 2017, 2018 and 2019, respectively. The ice melted on 29 June, 13 July,
and 29 July in 2017, 2018 and 2019, respectively. The ice thickness reached to a maximum of 1.62 m on May 29,
2017, 1.55 m on June 11, 2018, and 1.85 m on June 11, 2019 respectively.
A total of six Radarsat-2 scenes of Fine Quad-Pol mode (FQ13) were collected during December 2018-May 2019
(Table 1 and Figure 1). Each image was acquired in the ascending orbit and was quad-polarized (HH, HV, VH, and
VV). A total of 15 Sentinel-1 scenes of Interferometric Wide (IW) mode were collected during November 2018-April
2019. The IW datasets in descending orbit were dual-polarized (HH and HV). A total of six ALOS-2 scenes of HBQR
mode were acquired during January-April 2019. The HBQR datasets of the high-sensitive mode were also acquired
in the descending orbit and were quad-polarized (HH, HV, VH, and VV). InSAR coherence from HH polarization has
been proven to be higher and more stable than other polarizations in studying targets with low backscatter such as
permafrost (Chen et al., 2016; Samsonov et al., 2016) and some wetlands types (Chen et al., 2020). Preliminary studies
showed that HH polarization usually returned stronger backscatter which meant higher signal to noise ratio (SNR),



and was thus advantageous over other polarizations in monitoring low backscatter areas such as ice and water.
Therefore, HH polarization was chosen for ice monitoring in this study. The seasonal melting and freezing of ice
causes changes in surface temperature and salinity, so all images were acquired during December-May because ice
conditions were more stable. A Canadian Digital Elevation Data (CDED) of 30 m spatial resolution was used to
provide elevation information for image co-registration and topographic correction in InSAR processing. The
elevations in the study area including land and ocean range between 0 and 90 m above mean sea level according to
CDED.
The location of the Arctic Station building in Cambridge Bay was chosen as a reference for InSAR measurements
(RP). Four locations over the sea ice (P1, P2, P3, P4), representing different ice conditions, corresponding to water
depths of 10 m deep near harbor to more than 70 m deep in the strait (Canadian Hydrographic Service, 2007), were
selected to investigate the ice deformation using InSAR. Consistent coherence was also maintained at these locations
in interferograms from both ALOS-2 and Sentinel-1 images (Figure 3 and Figure 7). P1 was located in an area with a
water depth of 10-20 m, P2 in 40-60 m, P3 in 60-70 m, and P4 in 40-50 m (Figure 1). These locations were also 5-10
km away from the shore. No location near the harbor was selected for InSAR monitoring because of its small size and
proximity to land, making it easily affected by the resampling and filtering process. One location on the land near the
sea was chosen for studying the movements of the land (P5).
The along-track sea ice surface heights were available from the ATL07 product from NASA's second-generation
spaceborne lidar mission ICESat-2, with 17 m footprint, and 0.7 m along-track spacing. Each acquisition contains
profiles of sea surface/ice height segments from three beam-pairs. Each pair consists of a strong and a weak beam,
with pulse energies of the strong beams that are four times those of the weak (Kwok et al., 2019). In this study we
only used surface heights from three strong beam tracks at each acquisition time assuming that the measurements from
strong beams were more reliable than weak beams.

Table 1. SAR data used for the InSAR analysis.

| Sensor | Mode | Resolution (range x azimuth) | Incidence angles (degrees) | Repeat cycle (days) | Number of scenes acquired 2018-2019 |
|---|---|---|---|---|---|
| Radarsat-2 | FQ13 | 16.5 x 14.9 | 33 | 24 | 6 |
| ALOS-2 | HBQR | 4.3 x 5.1 | 39 | 14 | 6 |
| Sentinel-1 | IW | 2.7 x 22 | 33.4 | 12 | 15 |



**3 Methods**



In this study backscatter, coherence and deformation analyses were completed. First, SAR backscatter intensity of
different modes and sensors was compared with surface height variations from ICESat-2 data. Second, coherence
variations over ice from interferograms generated with different acquisition intervals on different months were
assessed. Third, the response of InSAR measurements to the surface height and ice draft was studied using time series
analysis.
Traditional differential InSAR (D-InSAR) procedures including co-registration, interferogram generation and phase
unwrapping were applied to each dataset with GAMMA software (Werner et al., 2000). First, co-registration was
applied to align and resample all images from one orbit (same sensor, polarization, and incidence angle) to a reference
image. Multi-looking and resampling were applied to reduce speckle effects and to improve coherence estimation.
Then, all possible InSAR pairs which were acquired within 90 days and perpendicular baseline less than 400 m were
used to generate differential interferograms. Atmospheric correction was also applied to each interferogram based on
the split-spectrum method which separates the ionospheric and the non-dispersive phase terms using spectral sub-band
images (Wegmueller, et al., 2018). The phase noise of an interferogram was reduced using an adaptive phase filter
(Goldstein and Werner, 1998), and then coherence and phase changes were estimated. The interferogram was
unwrapped to estimate the ice movements along the LOS using the Minimum Cost Flow algorithm (Costantini, 1998).
Backscatter variability can be used to study the ice types and conditions (Kwok and Cunningham, 1994; Casey et al.
2016). The backscattering coefficient was calculated to describe the interaction between incident microwaves with
sea ice. Its variation is related to wavelength, polarization, incidence angles, and ice properties such as surface
roughness, internal structure and dielectric properties. For the backscatter analysis, a radiometric calibration was
applied, and intensity values were converted to decibels (dB in $\sigma°$). Backscattering coefficients ($\sigma°$) were generated
along the ICESat-2 laser tracks to evaluate the SAR response to the surface height changes. During the original surface
height generation for ICESat-2 ATL07 product, the height segments were sampled to have overlap with each other
along the laser track. Hence, along-track height estimates were not independent because they included photons from
an earlier segment (Kwok et al., 2019). Surface height points also had higher spatial resolution than ALOS-2, Radarsat-
2 and Sentinel-1. Therefore, we generated the moving average in order to resample the surface height estimates from
ATL07 products to a resolution matching SAR data. $\sigma°$ values of SAR imagery and surface height measurements from
ICESat-2 acquired at the closest available dates were compared, assuming ice conditions remained the same between
the acquisitions. Correlation between SAR $\sigma°$ values and surface heights was calculated as well. By acquiring data
with similar incidence angles from ALOS-2, Radarsat-2 and Sentinel-1, backscatter differences in images at the same
time were expected to be largely related to the wavelength.
Once interferograms were generated, a threshold of 0.3 was used to identify coherent and non-coherent areas. Only
those areas which maintained high coherence (>0.3) were kept and correctly unwrapped for the next SBAS analysis.
Assuming no significant movement at RP (Figure 1) during the study period, the deformation change was set to zero
at RP. Then, a deformation map with respect to the reference pixel was computed to provide relevant information only
to those pixels with high coherence.



By exploiting all available SAR acquisitions, using interferograms generated from the D-InSAR technique with short
spatial and temporal baselines, the SBAS technique was applied to the to calculate the temporal evolution of
deformation. In the analysis, the Singular-Value Decomposition (SVD) algorithm was employed to achieve the
minimum-norm least squares solution and to extract meaningful deformation information at any time interval. Finally,
time-series cumulative deformation was achieved. In this analysis, negative values in InSAR deformation indicated
subsidence, while positive values in deformation indicated uplift. Then, cumulative ice draft growth was also
calculated from the measurements from the ice profiler and compared with cumulative deformations from InSAR
observations. Since time series surface heights were not available, the responses of InSAR deformation to the surface
height variation over a certain winter period were evaluated.


**4 Results**
**4.1 Observations of backscatter responding to surface height**
Data from the SAR sensors showed that the HH σ° values of landfast ice were generally low. For example, σ° values
of ALOS-2 and Sentinel-1 from P1-P4 varied in the range of (-29)-(-22) dB and (-35)-(-26) dB respectively, and only
varied within 5 dB at the same location during January-April, indicating that ice conditions were stable. Based on the
comparison of backscatter from three SAR sensors with surface height, backscatter from Sentinel-1 responded better
to surface height variations than ALOS-2 and Radarsat-2. The correlation between SAR σ° values and surface height
are given in Table 2.
For comparison with ALOS-2 backscatter, ground tracks from only two orbits of ICESat-2 were covered by ALOS-2
data. Two orbits were flown on 2019/1/30 and 2019/2/25 respectively. Scenes acquired on 2019/1/31 and 2019/2/28
were the two ALOS-2 data with the nearest dates corresponding to the ICESat-2 data. The σ° values of each ALOS-2
image in the HH polarization were compared with the surface height measurements along the ground tracks from three
strong beams of each orbit of ICESat-2 (Figure 3). Results showed that surface heights along the tracks of ICESat-2
acquired on 2019/01/30 showed less variation than that of 2019/2/25. SAR intensity from the HH polarization of L-
band ALOS-2 did not always show similar pattern as that of surface height (Figure 4). Ice surface heights varied in
the range (-3)-(3) cm along the tracks from all three strong beams of ISCESat-2, and SAR intensity from ALOS-2
acquired on 2019/1/31 varied in the range (-27)-(-19) dB along the same tracks (Figure 4A-C). In particular, both the
surface height and SAR intensity showed little variation along the track of 3L (Figure 4C). In contrast, ice surface
heights along the tracks of three beams of 2019/2/25 varied by (-3)-(16) cm, while SAR intensity from ALOS-2
acquired on 2019/2/28 varied in the range (-28)-(-18) dB (Figure 4D-F). The pattern of σ° values exhibited a greater
range and corresponded better to observed surface height values on 2019/2/25, which were also more variable (19
cm). Relative to 23 cm wavelength from ALOS-2, all measurements of surface roughness were well below a detectable
level, and therefore backscatter was low and not very responsive to the surface height changes. The correlation
between ALOS-2 σ° values and surface heights varied in each track of two dates with correlation coefficients range



from -0.1 to 0.78 (Table 2). Although track of 3L on 2019/1/31 provided the highest correlation coefficient (r=0.78),
the sample size was the smallest among six tracks.

Table 2. Correlation between ice surface height (from ICESat-2) and SAR backscattering coefficient ($\sigma°$ in dB) (*
only used the surface height larger than 0.3 m for correlation analysis).

| Date | ICESat-2 track | ALOS-2 | Radarsat-2 | Sentinel-1 |
|---|---|---|---|---|
| 2018-12-25 | 1R | NA | 0.07 | NA |
| | 2R | NA | -0.63 | NA |
| | 3R | NA | -0.1 | NA |
| 2019-01-23 | 1L * | NA | NA | 0.68 |
| | 2L * | NA | NA | 0.68 |
| | 3L | NA | NA | 0.54 |
| 2019-01-30 | 1L | 0.6 | NA | NA |
| | 2L | -0.1 | NA | NA |
| | 3L | 0.78 | NA | NA |
| 2019-02-25 | 1L | 0.2 | 0.23 | NA |
| | 2L | 0.68 | 0.03 | NA |
| | 3L | 0.51 | 0.5 | NA |
| 2019-02-28 | 1L | NA | NA | 0.83 |
| | 2L | NA | NA | 0.49 |
| | 3L | NA | NA | 0.55 |


For comparison with Radarsat-2 backscatter, ground tracks from only two orbits of ICESat-2 were covered by
Radarsat-2 data. Two ICESat-2 orbits were acquired on 2018/12/25 and 2019/2/25 respectively. Radarsat-2 FQ13
scenes acquired on 2018/12/22 and 2019/2/8 were the two datasets with the nearest dates corresponding to the ICESat-
2 data. The $\sigma°$ values of each Radarsat-2 image in the HH polarization were compared with surface heights along the
ground tracks from three strong beams of each orbit of ICESat-2 (Figure 5). Due to the relatively small footprint of
Radarsat-2, only a small area of overlap between Radarsat-2 and ICESat-2 was available for comparison. SAR
intensity from the HH polarization of C-band Radarsat-2 FQ13 did not show similar patterns to surface heights (Figure
6). Ice surface heights varied in the range 35-61 cm along the tracks acquired on 2018/12/25, and SAR intensity
acquired on 2018/12/22 varied in the range 6-13 dB along the same track. However, SAR backscatter fluctuated in
the opposite direction to surface height in many areas. Ice surface heights varied in the range (-2)-(17) cm along the
tracks acquired on 2019/02/25, and SAR intensity on 2019/02/8 varied in the range 2-13 dB along these tracks. In
general, for the ground tracks along the Radarsat-2 coverage, backscatter was not responsive to surface height changes.
The correlation between Radarsat-2 $\sigma°$ values and surface heights varied in each track of two dates with correlation



coefficients range from (-0.63) to (0.5) (Table 2). The sample numbers of all six tracks were very small, especially
along tracks of 1R and 3L. Radarsat-2 backscatter was much higher than ALOS-2 and Sentinel-1 images because ice
look up table (LUT) was applied to all Radarsat-2 scenes over the sea by default.
Sentinel-1 has very large coverage, so it covered more ICESat-2 tracks than ALOS-2 and Radarsat-2. σ° values of
Sentinel-1 IW of 2019/1/24 were compared with the surface heights of ICESat-2 of 2019/1/23 along the tracks of 1L,
2L and 3L, and σ° values of Sentinel-1 IW of 2019/3/1 were compared with the surface heights of ICESat-2 of
2019/2/28 along the tracks of 1L, 2L and 3L (Figure 7). Results indicated that the surface height and the SAR intensity
from Sentinel-1 along these tracks had large variations on both dates. For example, the surface height variation was
about 58 cm on the tracks of 2019/01/23 and about 45 cm on tracks of 2019/02/25. SAR backscatter appeared to be
highly correlated with ice surface height at many point locations (Figure 8). SAR backscatter was stronger in areas
where surface height was high, and SAR backscatter was weaker in the area when surface height was low. In particular,
ice surface heights varied in the range 0-58 cm along the tracks acquired on 2019/01/23, while SAR intensity acquired
on 2019/01/24 varied in the range (-35)-(-27) dB in the same areas (Figure 8A-C). Ice surface heights varied in the
range 31-76 cm along the tracks on 2019/02/28, and SAR intensity on 2019/3/1 varied in the range (-34)-(-27) dB
along the same tracks (Figure 8D-F). In general, surface heights along two ground tracks were as high as 80 cm above
sea level, and varied in range of 30-50 cm. The correlation between Sentinel-1 σ° values and surface heights were
generally strong in each track of two dates with correlation coefficients range from 0.49 to 0.83 (Table 2). The sample
numbers of all six tracks were very large, especially along tracks of 1R and 3L. It was found that σ° values may have
different patterns in places with surface height above 30 cm and below 30 cm. For example, the correlation between
σ° values and surface heights along track 1L on 2019/01/23 were 0.68 and 0.43 in places above 30 cm and below 30
cm respectively though the correlation was only -0.26 when combined.

## 278    4.2 Observations of coherence change over time

Coherence from C- band Radarsat-2 and Sentinel-1 and L-band ALOS-2 InSAR pairs were generated for the period
of December 2018 - May 2019. Coherence from interferograms generated over short and long acquisition intervals
was compared. In general, high coherence was observed over sea ice from L-band as early as in January, but high
coherence from C-band was only measured from mid-February. It was also found that coherence from L-band could
be maintained over a longer time period than C-band. For example, all InSAR interferogram pairs from ALOS-2 data
acquired from January to April 2019 produced good coherence. However, coherence was highest in interferograms
with a 14-day interval, and decreased gradually as time spans between acquisitions increased (i.e. from 28, 42, 56, 70
and 84-day intervals). Almost all ice areas remained coherent in the interferograms of 14 and 28-day intervals, while
only 60% of this area remained coherent in the interferograms with 84-day interval. Both the harbor and Queen Maud
Gulf area were coherent in most interferograms. With stable coherent ice areas, the SBAS analysis of time series
ALOS-2 interferograms was conducted for cumulative deformation estimates.



Radarsat-2 FQ13 coverage was smaller than ALOS-2 and Sentinel-1 coverage, thus the sea ice area observed was
limited to the Cambridge Bay harbor and nearby areas only. Coherence in interferograms of 48-day interval was
generally very poor compared to those of 24-day interval. Coherence values were low from interferograms generated
using images acquired during December and early February. It was observed that only harbor areas were coherent
from the interferograms of 2018/12/22_2019/1/15 and 2019/1/15_2019/2/8 among the images acquired before mid-
February. Coherence was greatly improved in the interferograms generated using images acquired between March
and mid-May. As such, the majority of ice areas were coherent from the interferograms of 2019/3/28/_2019/4/21 and
2019/4/21_2019/5/15/. SBAS analysis was not applied to Radarsat-2 data due to limited number of quality
interferograms.
Low coherence was observed in the interferograms generated from November Sentinel-1 images, and similar to
observations from Radarsat-2 images, coherence was also poor from December to January. Starting in January,
coherence was observed from ice in the middle of the Dease Strait but limited to small patches. Coherent areas were
also observed near Hurd Islands, an area beneath Dease Strait, protected from the influence of wind, waves and ocean
currents. Starting at mid-February, high coherence was observed across Dease Strait and area near Queen Maud Gulf
although there were still some areas including harbor with low coherence. From mid-March until the end of April, the
whole area including the harbor, Dease Strait and Queen Maud Gulf was highly coherent in Sentinel-1 12-day
interferograms, though half of the area lost coherence in interferograms with a 24-day interval. During this time high
coherence was observed in some areas over a 36-day period, though represented few, small and sporadically
distributed sites. Compared to the interferograms from ALOS-2 data, interferograms from Sentinel-1 before mid-
February were patchy and inconsistent. Therefore, only InSAR pairs from mid-February to April could be used for
the SBAS analysis.
Based on our observations, it was found that coherence over sea ice were strongly affected by temporal baselines. As
such, interferograms from longer time intervals, in particular from C-band Radarsat-2 data, may not be suitable for
ice deformation monitoring due to low coherence, and a small temporal baseline subset is preferred for ice deformation
estimates.

**4.3 Characterizing the response of InSAR deformation to ice draft growth**
The cumulative estimates of deformation in the LOS direction from SBAS analysis of both ALOS-2 and Sentinel-1 at
four locations were compared with ice draft measurements. No cumulative estimates were available from Radarsat-2
due to a lack of sufficient interferograms for SBAS analysis, instead, differential estimates of deformation at only two
locations were compared with ice draft. All InSAR measurements were calibrated as changes relative to the RP.
Results from ALOS-2 (Figure 9A) during January-April 2019 and results from Sentinel-1 (Figure 9B) during
February-April 2019 both indicated that the land location P5 experienced only a slight fluctuation in position during
the study period. InSAR results from both sensors indicated that the general subsidence trends in deformation were
the same at P1 and P4 locations (Figure 9). More than 30 cm deformation at P4 was detected from both ALOS-2 and



Sentinel-1 observations. However, the deformation trends measured from two sensors were opposite at P2 and P3
locations.
According to the ice profiler at the dock location of Cambridge Bay, ice draft began to accumulate from 2018/10/13,
became 40 cm thick by the end of October, and peaked at 1.85 m on 2019/6/8, then decreased until it was completely
melted on 2019/7/2 (Figure 10). Results indicated that ALOS-2 InSAR observations at four locations showed different
movements as a response to a 62 cm ice growth during January 17-April 11, 2019, the ALOS-2 observation period.
For example, cumulative deformation from ALOS-2 InSAR indicated an uplift of 57 and 52 cm at P2 and P3, but a
subsidence of 6 and 31 cm at P1 and P4 (Figure 9A). In particular, a downward deformation of about 24 cm occurred
during February 28-March 28 at P4. At the same time, Sentinel-1 InSAR observation at all four locations showed a
trend of subsidence in response to an increase of 30 cm of ice draft during February 17-April 6, 2019, the Sentinel-1
observation period. For example, cumulative deformation from Sentinel-1 InSAR indicated a downward motion of
2.5, 18, 8 and 32 cm at P1, P2, P3 and P4, respectively (Figure 9B). It was observed that ice at the P3 location
experienced a downward movement during March 1-March 25, then an uplift movement during March 25-April 6,
2019. D-InSAR results from three pairs of Radarsat-2 acquired from February to May measured a cumulative uplift
of 7 cm at P1, but a subsidence of 13 cm at P4 in response to the ice draft growth of 53 cm during the same time period
(Table 3). The subsidence trend at P4 detected by Radarsat-2 was consistent to that observed from ALOS-2 and
Sentinel-1 data. However, the deformation magnitude detected from Radarsat-2 was much less than that from ALOS-
2 and Sentinel-1.

Table 3. D-InSAR measurements at location P1 and P4 using Radarsat-2 FQ13 data vs ice draft measurements from
the ice profiler located close to RF.

|  | InSAR (P1) | InSAR (P4) | Ice Draft |
|---|---|---|---|
| 2019/02/08-2019/03/28 | 0.02 | -0.01 | 0.33 |
| 2019/03/28-2019/04/21 | 0.03 | -0.04 | 0.12 |
| 2019/04/21-2019/05/15 | 0.02 | -0.08 | 0.08 |


**4.4 Characterizing the response of InSAR deformation to surface height variation**
In order to understand how InSAR derived measurements of deformation corresponded to surface height changes, ice
surface height changes derived from ICESat-2 during the SAR observation period were analyzed. Although ICESat-
2's repeat cycle is 91 days, no two acquisitions from the same orbit were available within the 91-day cycle, however,
there were two acquisitions from the same path for a 182-day interval. The same ground track was scanned on
2018/10/24 and 2019/4/24 individually. Another track in the vicinity of these two acquisitions, with a distance of 345
m, was acquired on 2019/1/23. However, among three strong beams from each acquisition, only two beams from the
acquisition of 2019/1/23 were close to the two beams of the acquisition of 2018/10/24 and 2019/4/24. Therefore, the





surface height variations from these three dates along two ground tracks were compared. In the analysis, line 1R of
2018/10/24, line 2L of 2019/01/23 and 1L of 2019/04/24 were simply considered as one ground track spatially for
comparison (referred to as Line 1) (Figure 11 A-C). Line 2R of 2018/10/24, line 3L of 2019/01/23 and 2L of
2019/04/24 were considered as the other ground track spatially for comparison (referred to as Line 2) (Figure 11 D-
F).
Surface heights were plotted against latitude for comparison as the general flight direction was north to south. It was
found that surface heights along Line 1 and Line 2 were generally below mean sea level on 2018/10/24, mostly in the
range of (-25)-(-10) cm (Figure 11A for Line 1 and Figure 11D for Line 2), then increased above sea level on
2019/01/23, mostly in the range of 30-60 cm (Figure 11B for Line 1 and Figure 11E for Line 2), and remained above
sea level on 2019/04/24, mostly in the range of 5-40 cm (Figure 11C for Line 1 and Figure 11F for Line 2). The
variations of heights in three dates from these two tracks suggested that the surface height was negative or below sea
level at the beginning of ice forming, then increased to at least 30 cm above sea level in mid-January, then decreased
to some extent in April, but remained at least 5 cm above sea level. However, surface heights showed much greater
variations at different locations along the track over a short distance in January and April than in October.
InSAR deformation is the measurement of change along the LOS projection. It contains components of both horizontal
and vertical movements of sea ice. Line 1 and Line 2 from ICESat-2 were located far from Cambridge Bay, they were
only covered by Sentinel-1 imagery. Due to the small coverage, no pairs of the same or close orbit from an ICESat-2
acquisition were covered by ALOS-2 and Radarsat-2 during the study period. Therefore, the InSAR deformation
measurements from ALOS-2 and Radarsat-2 were not compared with surface height changes. Cumulative deformation
estimates from Sentinel-1 images during 2019/2/17-2019/4/6 were compared with the ICESat-2 surface height
changes from 2019/1/23 and 2019/4/24 (Figure 12). For this comparison, the surface height changes at each sample
point between two close orbits along Line 1 and Line 2 were calculated. Specifically, the surface height differences
of Line 1 were calculated by subtracting the heights of 2L of 2019/1/23 from that of 1L from 2019/4/24 (the closest
ground tracks during the ICESat-2 flight at two dates). Similarly, the surface height differences of Line 2 were
calculated by subtracting the heights of 3L of 2019/1/23 from that of 2L from 2019/4/24. Moving average of height
measurements at a distance similar to the resolution of Sentinel-1 SBAS product of 40 m was used. Similar patterns
of surface height changes were found in Line 1 and Line 2. InSAR measurements in most locations along track Line
1 consistently measured subsidence which was consistent with a decrease in surface heights, however, at locations
between latitude 68.67-68.7 degrees, the InSAR measurements indicated a slight uplift. It was found that the
magnitudes of InSAR deformation along these two tracks were smaller than surface height changes. For example,
surface height changes ranged from (-35)-(-13) cm, but InSAR only reflected the changes that ranged from (-8)-1 cm
along the same track.

**5 Discussion**





There is limited information that can be used to validate InSAR measurements of ice deformation in the Arctic ocean.
In this study, we investigated how InSAR deformation measurements from L-band ALOS-2, and C-band Radarsat-2
and Sentinel-1 corresponded with the ice draft measured from an in-situ ice profiler, as well as surface heights
measured from ICESat-2. We also examined the response of backscatter variations to surface height changes along
the ICESat-2 ground tracks and evaluated the coherence changes during the winter season. Deformation magnitudes
in some locations from ALOS-2 and Sentinel-1 were found to be in the same range of ice draft growth measured from
one location. Spatial and temporal patterns of Sentinel-1 SAR signal were generally consistent with surface height
changes. However, variations between the InSAR observations and measurements from ice profiler and ICESat-2
remained. The variations may be attributed to many factors including sensor characteristics (e.g., wavelength, repeat
cycle), and ice surface conditions (e.g., snow cover, salinity, surface height, thickness, roughness, stability, types, air
bubble concentration, and deformation) (Kwok and Cunningham, 1994; Eriksson et al., 2010; Dierking et al., 2017;
Cafarella et al., 2019) in addition to coherence loss from temporal decorrelation.
Results showed that the temporal variability of backscatter from landfast ice (first-year) was relatively low in
Cambridge Bay. Low variability of backscatter has also been observed from landfast ice (multi-year) in Beaufort Sea
in winter (Kwok and Cunningham, 1994; Steffen and Heinrichs, 2001). Throughout most time of the winter, sea ice
near Cambridge Bay is covered by snow. Given the fact that the snow depth in Cambridge Bay only reached to a
maximum of 8 cm during April-May (Howell et al., 2016), it is not likely to have had a significant influence on
backscatter, especially at C- and L-band backscatter due to their penetration capacity (Kwok and Cunningham, 1994;
Kwok, 2010).
Our analysis indicated that temporal changes in Sentinel-1 backscatter values corresponded to changes in surface
heights measured along two ground tracks of ICESat-2. In some cases, ice surface height measured from ICESat-2
can have uncertainties due to the snow cover (Kwok, 2010), however this is not expected to have had a significant
impact here due to low snow depth. In previous studies, backscatter from C-band Radarsat-2 and L-band ALOS-2
were found to be correlated to the FYI surface roughness measured from airborne laser scanner, and backscatter
increased with larger surface roughness (Cafarella et al., 2019). Low surface heights from ICESat-2 were reported to
correspond to low backscatter in ice leads and floes (Kwok et al., 2019). In this study, it was found that variation of
SAR backscatter from Sentinel-1 corresponded to changes of surface height from 0-76 cm, as generally, higher ice
surfaces corresponded to the higher SAR backscatter. However, such a pattern was not obvious between backscatter
from C-band Radarsat-2. With very similar incidence angles from both C-band SAR sensors (Table 1), similar results
were anticipated. Longer acquisition interval between Radarsat-2 and ICESat-2 may have contributed to the low
correlation. Comparing to only 1-2 days of difference in acquisitions between ICESat-2 and SAR data from ALOS-2
and Sentinel-2, there were 17 days difference in ICESat-2 and Radarsat-2 acquisition dates. As such, ice conditions
may have changed during such period. We believe the low responses to surface height changes from the backscatter
of ALOS-2 data and Radarsat-2 data could also be due to the low surface height and less variation along the available
tracks covered by data from those two sensors. Especially, for ALOS-2 data with a 23 cm wavelength, surface height
variations less than 10 cm would not have impacted backscatter. Thus, it is likely that backscatter over ice from L-





band data may not correspond to the true ice surface due to strong penetration into the ice and snow (Dierking et al.,
2017). Only a small number of samples along the surface height track were used for comparison to the Radarsat-2
data due to their small coverage, which might have also contributed to these results. Further correlation analysis of
backscatter correspondence with surface height is needed.
The pattern and density of interferogram fringes describe the deformation type and rate, with frequent fringes
indicating a higher density of phase changes and greater deformation. Generally, higher fringe density indicates
unstable ice conditions (Dammann et al., 2019), and parallel fringes indicate strong lateral movement. Parallel and
dense fringes generated over the majority ice area before February in this study therefore indicated that strong lateral
deformation was dominant, which corresponded to the rapid ice growth observed from the ice draft measurement
during October-January (Figure 10). The existence of both the parallel and circular fringes from time series of
interferograms from three sensors indicated that there were both lateral and vertical motion over the landfast ice in the
study area. Circular and less dense fringes were observed over the majority of the ice area starting in February (from
January with ALOS-2) also indicating that vertical deformation (uplift or depression) was dominant. Decreasing fringe
density as ice thickens over the winter has been reported before (Dammann et al., 2016), while others have also used
fringe densities in general to study the stability conditions of ice (Dammann et al., 2016; Dammann et al., 2019). In
those studies, dense fringe areas were considered unstable. One of the differences between ALOS-2 interferograms
and Sentinel-1 interferograms was the fringes. In this study, ALOS-2 interferograms from January were considered
reliable for deformation analysis, however, Sentinel-1 interferograms were only reliable starting from mid-February.
Sentinel-1 InSAR interferograms from data collected before February showed parallel and dense fringes. Both ice
draft and surface height are measures of vertical changes of ice conditions, therefore, assuming that InSAR
measurements were from stable ice, this mainly corresponded to vertical deformation and could be related to vertical
height changes of landfast ice. However, InSAR phase near the dock where the ice profiler located measured little
change, meaning that the ice was frozen to the sea floor. As such, no direct link between the InSAR measurement and
ice draft at this location was made. Sentinel-1 interferograms generated from images acquired before mid-February
were not used to provide information related to vertical height changes.
Temporal decorrelation over ice was observed from C-band Radarsat-2 and Sentinel-1 in this study. Coherence from
C-band only lasted within a 24-day interval compared to L-band ALOS-2 data, which maintained coherence over an
84-day interval. This can be explained by the penetration depth of different wavelengths. With a stronger penetration
depth at L-band, the SAR signal is more sensitive to internal ice structures which are more stable in landfast sea ice
compared to C-band which is more sensitive to its surface structures (Howell et al., 2018). It was found that the
coherence from all sensors over ice was high near the harbor and along the coast where ice attached to the coastline
or sea floor, then decreased toward the open ocean. InSAR coherence was generally poor from the images acquired
before January. Similar observations of lower correlation over sea ice from C- and X- band compared to L-band were
also reported in other studies (Meyer et al., 2011; Dammann et al., 2018). Poor coherence over ice before January was
probably caused by unstable ice condition and mobile ice. The coherence was lost when the magnitude of movement
was large relative to the SAR wavelength and the shift between the two acquisitions changed the ice surface,
consequently, the interferograms could not be generated using C-band data before January.



Some local uplift and subsidence were observed in both SBAS analysis of ALOS-2 and Sentinel-1 InSAR data. As
InSAR only measures the LOS component of the three-dimensional deformation, actual ice deformation may not be
well represented in the study area. Also, deformation trends from one dataset may not be consistent between datasets.
For example, at P2 and P4 uplift was detected from ALOS-2, and as subsidence from Sentinel-1, although similar
subsidence trends were detected at two other locations at P1 and P4 from both ALOS-2 and Sentinel-1. Such a
difference may be caused by sensitivity from different wavelengths and differences in acquisition dates. In general,
the deformation measured from both ALOS-2 and Sentinel-1 was found to be within the same magnitude and range
of ice draft growth. For example, a maximum deformation of 57 cm was detected from ALOS-2 corresponding to a
62 cm ice draft growth during the same period. Similarly, a maximum deformation of 32 cm was detected from
Sentinel-1 corresponding to 30 cm ice draft growth during the same period. Unfortunately, no in-situ deformation
information was available to validate the InSAR results at different locations.
Landfast ice can often grow and cover a large area (Dammann et al., 2019). The results from this study showed that
the harbor area had near-zero phase change during the winter because of the ice anchored to the floor, which
distinguished it from the area further into the ocean. Large areas with coherence and various fringe density in the study
area also demonstrated the possibility of ice extent and stability mapping using InSAR which has been applied
elsewhere (Meyer et al., 2011; Dammann et al., 2018; Dammann et al., 2019). The pattern and density of the fringes
may be used to infer different stages of landfast ice. With the approach demonstrated in this paper, observations from
frequent SAR acquisitions showed promise for more reliable and consistent results with the in-situ measurements and
other available products. Studying ice stability and deformation using InSAR may be a viable solution for the Arctic
communities.

**6 Conclusions**
In this paper, the potential of InSAR to observe landfast ice deformation in the Arctic ocean was investigated using
C-band Radarsat-2 and Sentinel-1 and L-band ALOS-2 satellite data. SBAS approach was applied to generate the time
series deformation information. InSAR measurements were compared with both the ice draft measured from an in-
situ ice profiler and surface height measured from ICESat-2 satellite in order to understand the response of vertical
ice deformation to ice thickness changes. Circular fringes were observed over sea ice during January-April and may
represent vertical deformation corresponding to steady ice growth. Results indicated that uplift and subsidence patterns
observed from ALOS-2 varied in different locations, and the magnitudes of deformation were in the range (-31)-57
cm comparing to 62 cm ice draft growth during the same period. Subsidence observed from Sentinel-1 were in the
range (-32)-(-2.5) cm comparing to 30 cm ice draft growth. The trends of InSAR deformation from Sentinel-1 were
also consistent with the ice surface height changes detected from ICESat-2 along two ground tracks. L-band ALOS-2
achieved higher coherence over longer acquisition intervals compared to C-band Radarsat-2 and Sentinel-1 due to
temporal decorrelation. The challenge of temporal coherence loss could be resolved with higher temporal revisits of
new modern SAR sensors such as the RADARSAT Constellation Mission launched in June of 2019. SAR backscatter



from Sentinel-1 was found to be sensitive to the surface height changes along the ground track of ICESat-2 with
correlation coefficients in the range 0.49-0.83. However, correlations between SAR backscatter and surface height
varied and were not strong from the Radarsat-2 and ALOS-2 possibly due to limited data samples covering area with
little surface roughness. Results indicated that InSAR data with a high temporal revisit shows promise for investigating
the Arctic sea ice deformation over a large area; however, more data is needed to validate the satellite monitoring of
sea ice thickness changes.

**Data availability.** Sentinel-1 data can be obtained free of charge from the Copernicus Open Access Hub
(https://scihub.copernicus.eu/, last access: 2 October 2019). Ice draft data from Cambridge Bay, Nunavut, can be
obtained from Ocean Networks Canada (http://www.oceannetworks.ca, last access: 1 December 2019). ICESat-2 data
can be obtained from National Snow and Ice Data Center (https://nsidc.org/data/icesat-2, last access: 10 September
2019). The ALOS-2 data were obtained under a scientific license (see Acknowledgements) for an approved proposal
submitted to JAXA. The Radarsat-2 data were obtained from Canadian Space Agency. Both ALOS-2 and Radarsat-2
data are not publicly accessible.

**Author contributions.** ZC conducted the interferometric processing and analysis and drafted the initial manuscript.
BM provided critical expertise related to sea ice interpretation and analysis. JD and JP managed the project. All co-
authors provided valuable recommendations, editing and corrections for the final manuscript.

**Competing interests.** The authors declare no conflict of interest.

**Acknowledgements**

ALOS-2 imagery was provided by Japan Aerospace Exploration Agency under the "2nd Earth observation research
announcement collaborative research agreement (non-funded)". Radarsat-2 imagery was provided by Canadian Space
Agency. We thank the European Space Agency for access to the Sentinel-1 data, the Ocean Networks Canada for
sharing the ice draft data, NASA for access to the ICESat-2 data. RADARSAT-2 Data and Products © Maxar
Technologies Ltd. (2018) – All Rights Reserved. RADARSAT is an official mark of the Canadian Space Agency.

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

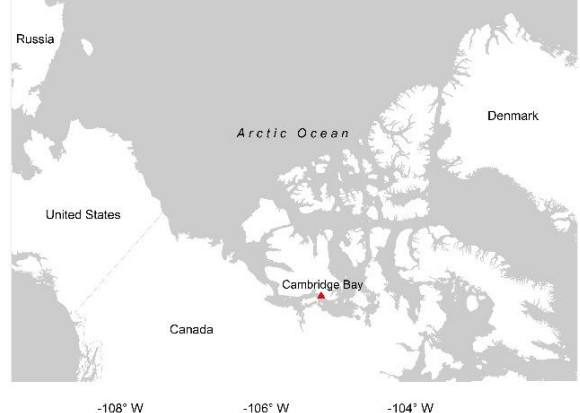


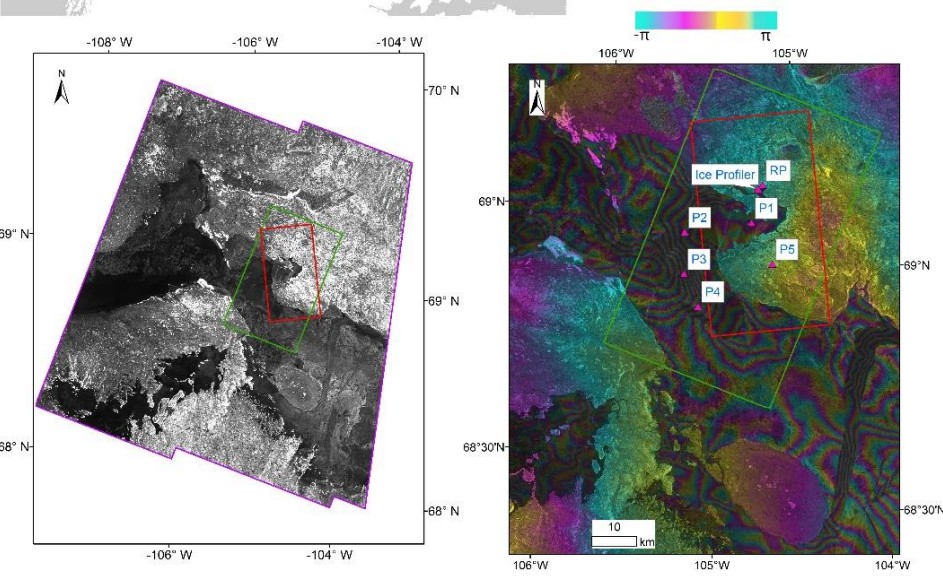

Figure 1. Upper left: map of study area showing the Arctic and Cambridge Bay location (the red triangle). Lower left:
Cambridge Bay study area covered by Sentinel-1 IW mode (purple line), ALOS-2 HBQR mode (green line) and
Radarsat-2 FQ13 mode (red line). The background is a Sentinel-1 image acquired on 2019/3/25. Right: locations of





Ice Profiler and InSAR measurement (RP, P1, P2, P3, P4, P5). The background is an example interferogram with clear
fringes over the sea ice generated using Sentinel-1 IW mode of HH polarization acquired on 2019/3/25 and 2019/4/6.

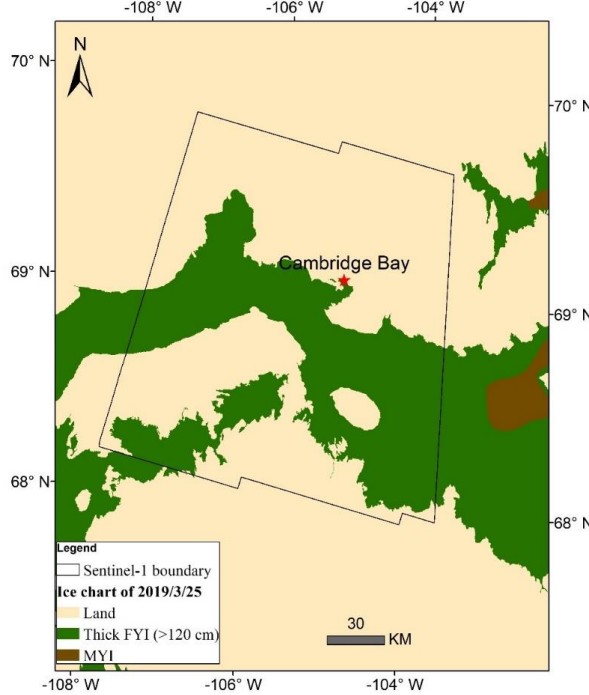


Figure 2: Regional ice chart for the study area provided by the Canadian Ice Service Data Archive (CISDA). Chart is
valid for March 25, 2019.







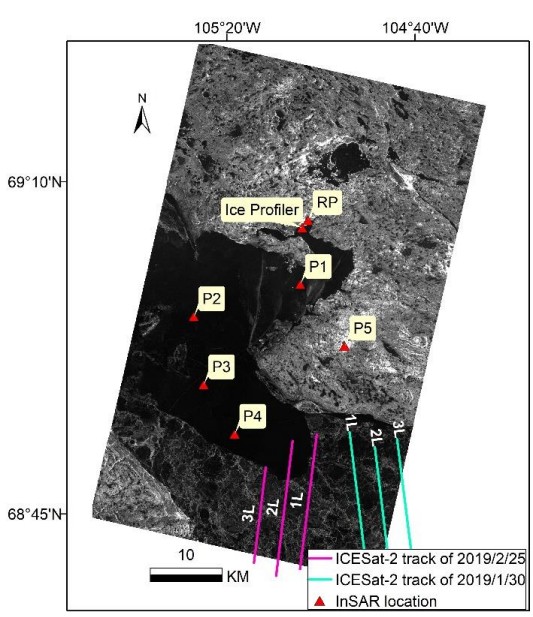

Figure 3. Locations of Ice Profiler, InSAR measurement (RP, P1, P2, P3, P4, P5) and ground tracks of ICESat-2 on a
HH polarized ALOS-2 image from 2019/2/28.





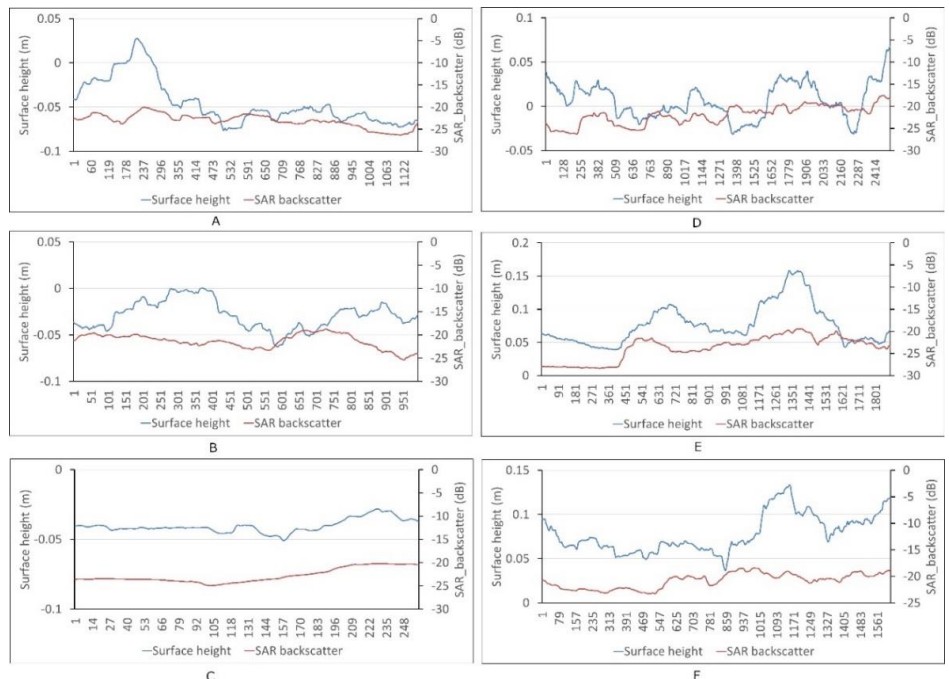

Figure 4. Comparison of surface height measurements from ICESat-2 with SAR backscatter from ALOS-2 (Red line shows the SAR backscatter and blue line for the surface height). Left: ICESat-2 from 2019/01/30 vs ALOS-2 from 2019/01/31 (values recorded from South to North along the track) ((A), (B) and (C) corresponds to the ground track of 1L, 2L and 3L); Right: ICESat-2 from 2019/02/25 vs ALOS-2 from 2019/02/28 (values recorded from North to South along the track) (D, E and F corresponds to the ground track of 1L, 2L and 3L).



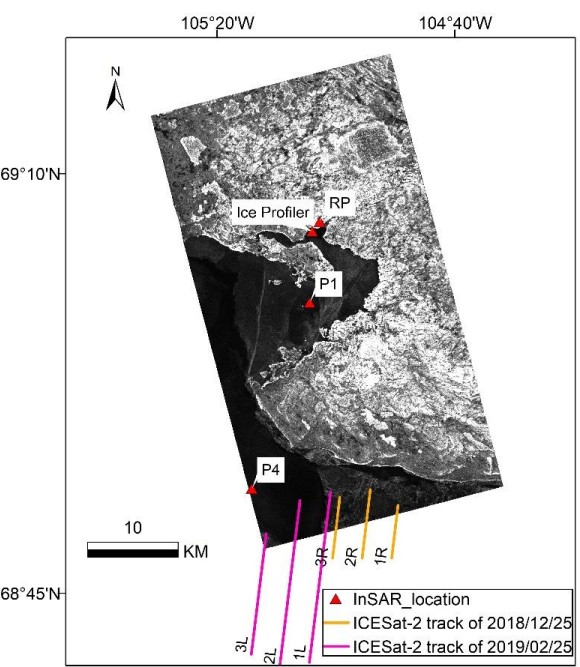


Figure 5. Locations of Ice Profiler, InSAR measurement (RP, P1, P4) and ground tracks of ICESat-2 on an HH
polarized Radarsat-2 FQ13 image from 2019/2/8.





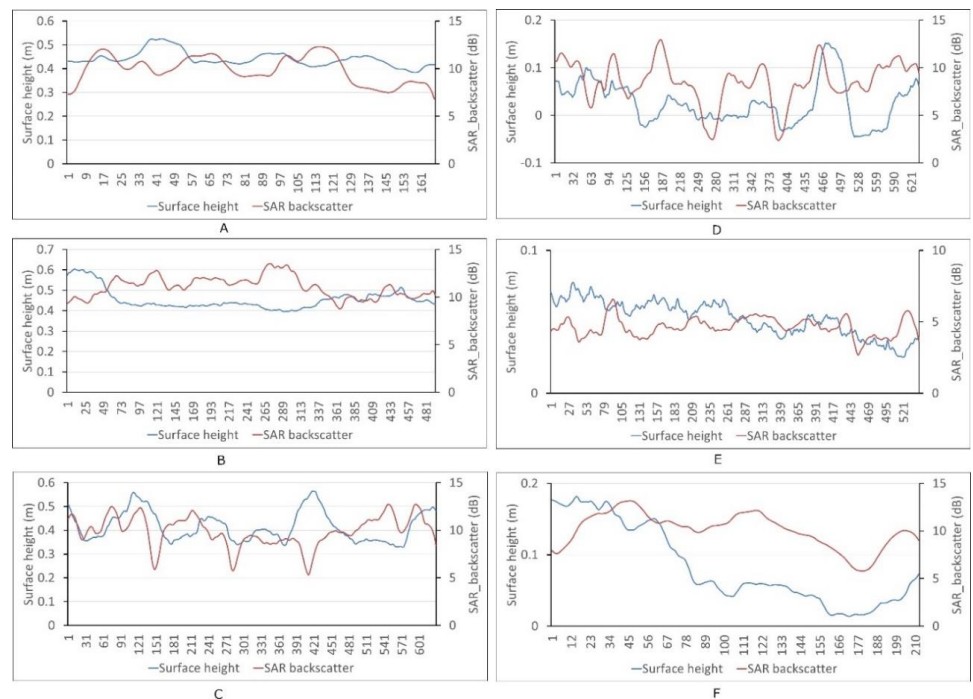


Figure 6. Comparison of surface height measurements from ICESat-2 with SAR backscatter from Radarsat-2 FQ13
(Red line shows the SAR backscatter and blue line for the surface height). Left (A-C): ICESat-2 from 2018/12/25 vs
FQ13 from 2018/12/22 (values recorded from North to South along the track); Right (D-F): ICESat-2 from 2019/02/25
vs FQ13 from 2019/02/08 (values recorded from North to South along the track); (A-C). the ground track of 1R-3R;
(D-F). the ground track of 1L-3L.





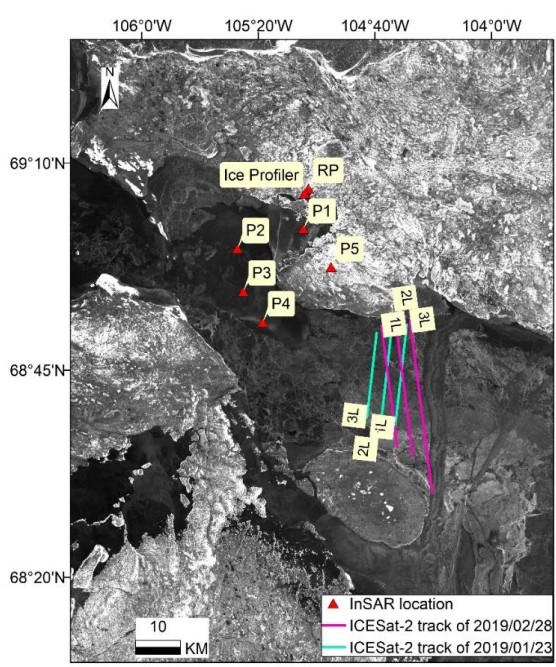


Figure 7. Locations of Ice Profiler, InSAR measurement (RP, P1, P2, P3, P4, P5) and ground tracks of ICESat-2 on a

HH polarized Sentinel-1 IW image from 2019/3/1.



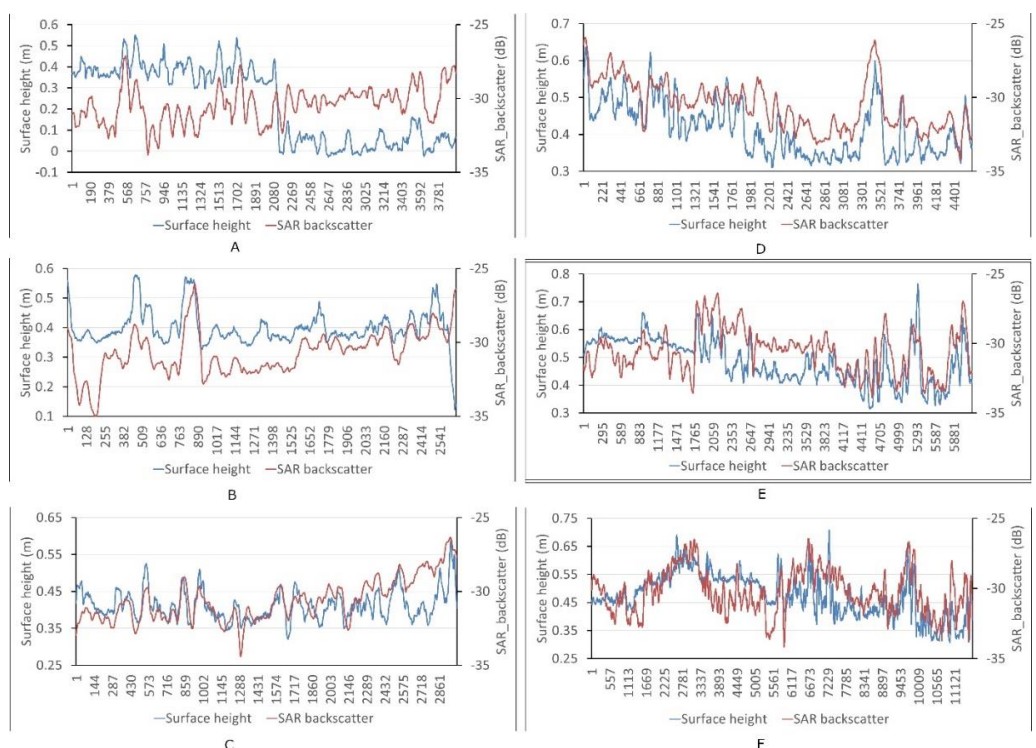

Figure 8. Comparison of surface height measurements from ICESat-2 with SAR backscatter from Sentinel-1 IW (Red
line shows the SAR backscatter and blue line for the surface height). Left (A-C): ICESat-2 on 2019/01/23 vs IW on
2019/01/24 (values recorded from North to South along the track); Right (E-F): ICESat-2 on 2019/02/28 vs IW on
2019/03/01 (values recorded from South to North along the track); (A-C). the ground track of 1L-3L; (D-F). the ground
track of 1L-3L.



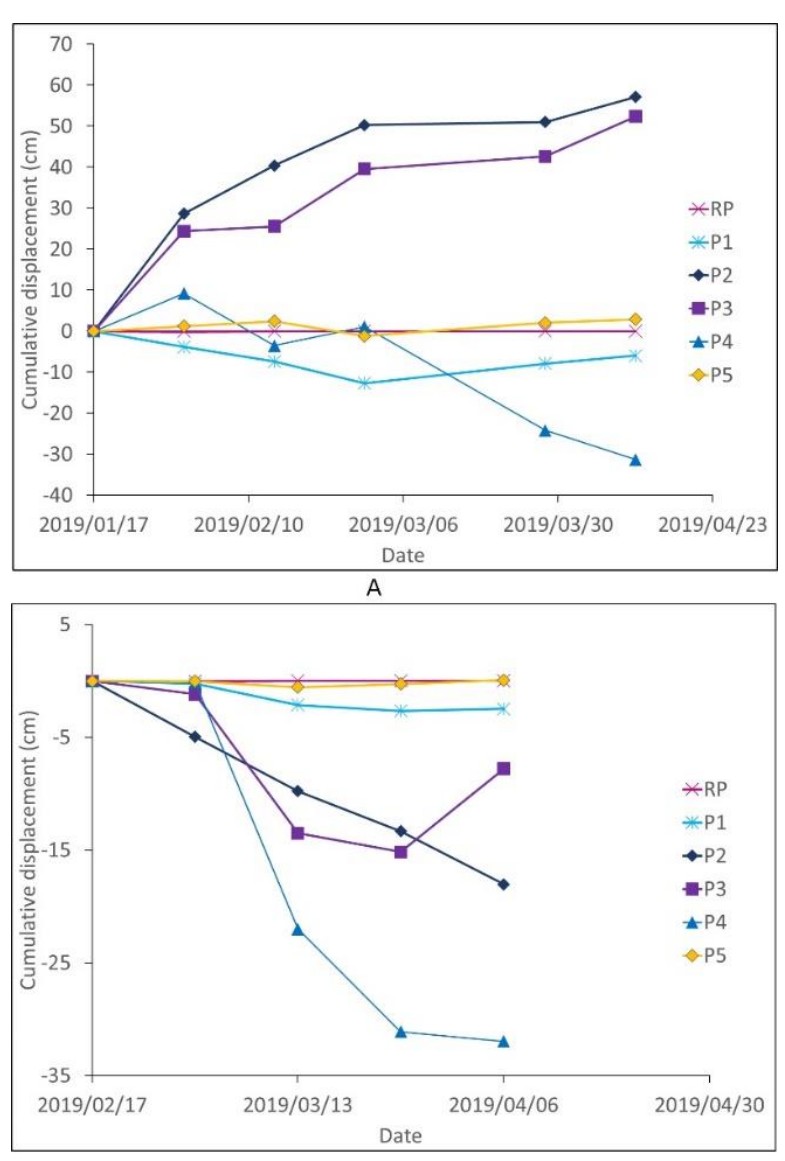


Figure 9. (A). ALOS-2 InSAR cumulative deformation. (B). Sentinel-1 InSAR cumulative deformation. RP is a
reference location. P1-P4 are locations over ice, and P5 is a location over land.




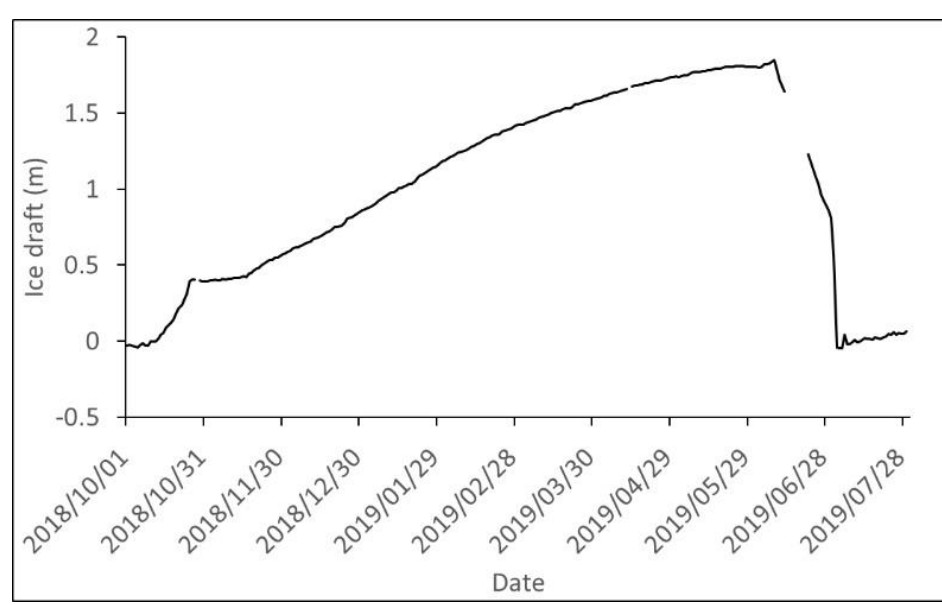


Figure 10. Ice draft changes from a ice profiler located at harbor of Cambridge Bay.





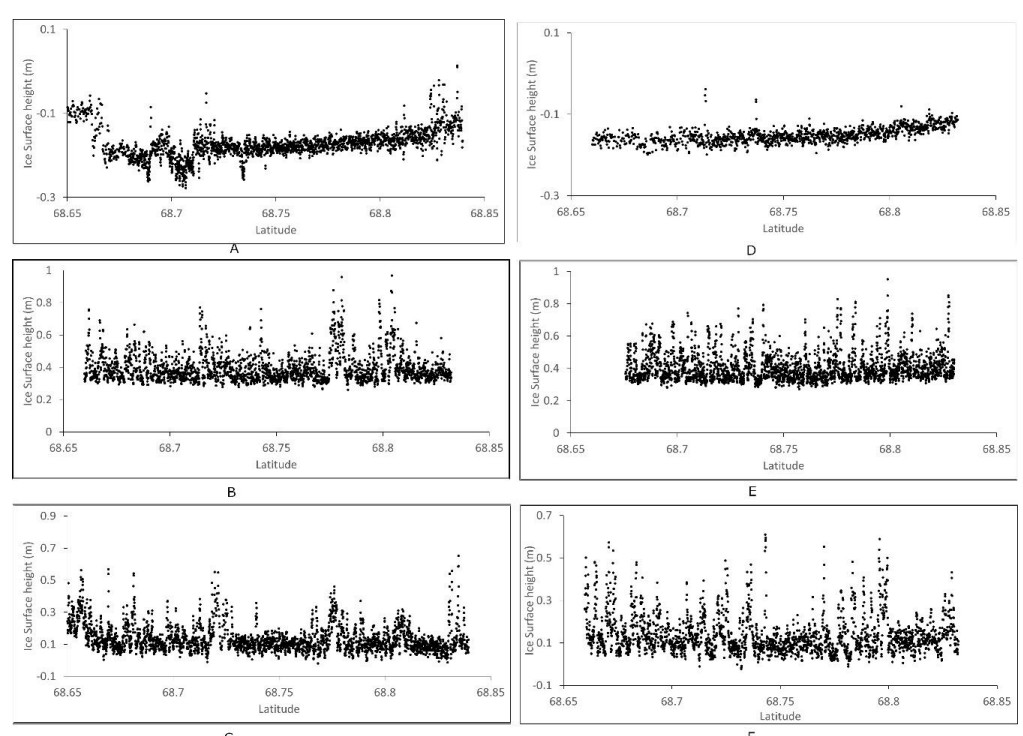

Figure 11. Comparison of surface height measurements from ICESat-2. (A) 1R of 2018/10/24, (B) 2L of 2019/01/23, (C) 1L of 2019/04/24; (D) 2R of 2018/10/24, (E) 3L of 2019/01/23, (F) 2L of 2019/04/24



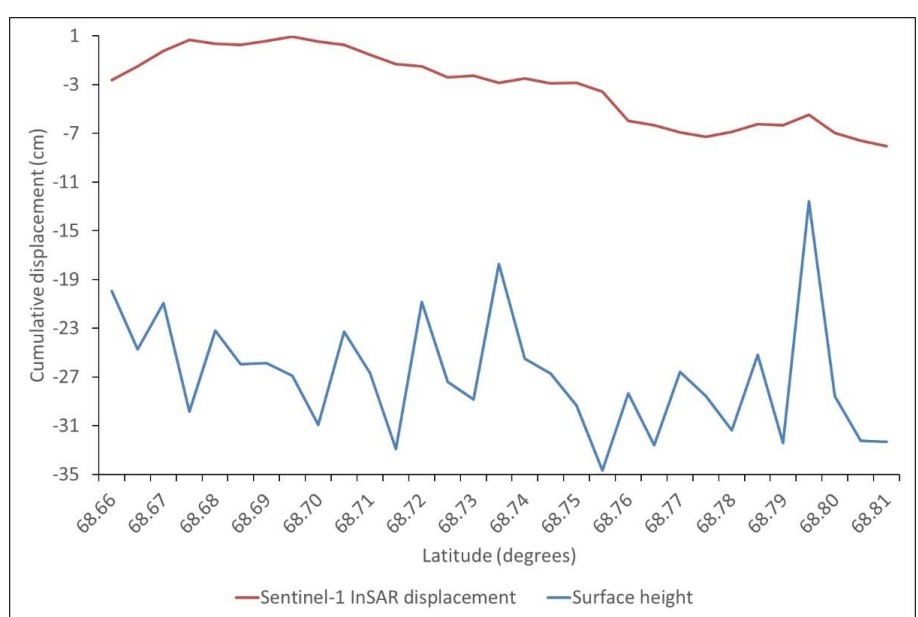


Figure 12. Comparison between Sentinel-1 InSAR deformation and surface height changes measured from ICESat-2
during January-April 2019 along Line 1 track.