# Peer review of "InSAR monitoring of Arctic land fast sea ice deformation using"

_The Cryosphere, 2021_

## Referee Comment (RC1)

**Review**

InSAR monitoring of Arctic land fast sea ice deformation using L-band ALOS-2, C-band
Radarsat-2 and Sentinel-1

By Chen et al.

This paper presents a study comparing SAR data from multiple platforms including PALSAR-2,
Radarsat-2, and Sentinel-1. The analysis incorporates and compares SAR-based backscatter,
coherence, and interferometric analysis with IceSat-2 freeboard measurements and in-situ ice
profiler data. The comparison and utilization of these datasets is timely as the idea of comparing
InSAR measurements with ice thickness/roughness is interesting. The authors have obviously put
a lot of time into this research so it is unfortunate to see that the analysis and interpretation is
both confusing and based on a series misconceptions. I recommend that the entire analysis and
interpretation must be changed.

**General comments on the results sections**

**Section 4.1**
First of all, the comparison suggests that Sentinel-1 matches surface roughness better than ALOS
and Radarsat. However, the comparison is made predominately on different areas and different
level of roughness. It happens that the best comparison is over the roughest area, not surprising
and uninteresting. The paper also lacks a proper discussion into the different sensitivities based
on L- and C-band.

Second, it is not clear what this analysis has to do with InSAR. The incorporation of this should
be made more clear in the introduction and title. The section is somewhat rambling going back
and forth between dates and dB, which is difficult to follow. All of this could be summed up with
a table with statistical measurements like RMS error etc. Furthermore, the description of what
data overlaps should be included in the data section. Also, the figures could be annotated better
to make the comparison and linkages to the text clearer.

**Section 4.2**
This section is quite uninteresting as all of this is known and expected - large baseline, lower
coherence. All of this text could be replaced by a short paragraph, which you sort of include in
the end at line 311. If you want to include coherence detail, this should be done through
coherence images so the reader can view for themselves. Furthermore, "high" and "poor" and
"low" coherence is all relative and is not particularly helpful. I suggest taking out this whole
section or include some details in the data section.

**Section 4.3**
It is not clear how you separate vertical shifts from horizontal motion. From Figure 9, it looks
like you have plenty of horizontal motion. Why is this not discussed? You also show only one
interferogram in the whole paper. You should give the reader a chance to look at fringe changes
and at least the difference between the sensors.

**Section 4.4**

The results here start to get at something interesting. What the results show is that over the IceSat-2 track, thinner ice may be associated with more deformation. This is expected, but a nice result. However, there are many other things that come into play e.g. distance from shore, forcing conditions etc. that would need to be discussed.

However, it is unclear where the authors take this analysis as they seem to misunderstand what this information tells them. It seems like the authors think that the ice movement is not lateral sea ice movement, but rather vertical ice displacement. If that is the case, it makes little sense to me to compare spatio-temporal change in the InSAR transect with just spatial change in ice elevation. Furthermore, line 489 points to a serious misconception as steady ice growth should not result in fringes at all. In fact, in this area, ice growth may occur fairly homogenous, in which case any ice growth would not result in fringes. I think most of the interferometric signal you see is due to horizontal ice motion. This is also why you have opposite responses in Figure 9.

Line 491 points to another serious misconception. Ice growth of 62 cm should lead to an average of 6 cm uplift of the freeboard, nowhere near 57 cm.

**Additional general comments**

Writing should be improved as there are long sentences, comma errors which interrupts the flow. Also, be careful of terms and consistency. Avoid passive voice.

The manuscript should be condensed in all sections. Also consider including subsections in the data, methods, and discussion as you are dealing with a lot of different data and methods. For instance, Section 2 could be split into ice profiler, SAR data etc. Here, discuss accuracy of ice profilers, where it is located etc. Figure 10 should go in that section. Same for Section 3, you should also have a subsection where you describe your InSAR workflow, lidar processing etc.

**Detailed comments:**

Title: Landfast is one word

Abstract: Line 11: reformulate so it doesn't seem like you measure draft with SAR.

Line 15: minus and hyphen should be different or write "to"

Line 64: What do you mean by, "it is difficult to verify"? Do you think it should need to be verified more? InSAR-based deformation measurements have been validated previously using DGPS sensors in Alaska, but has not been cited here.

Line 91: "Understanding the response…." This sentence is not clear to me. It seems out of the blue and lots of terms that should be discussed more if you bring up here e.g. sources of deformation, thickness change below sea level (do you mean ice growth), changes in surface height, do you mean ridging or snow? etc.

Line 98: Here it is also unclear how the validation will improve these products. Please specify.

Line 100: "may provide the information with increased spatial and temporal coverage…" This sentence should also be clarified further I can't remember if you brought up spatial and temporal coverage before regarding ice profilers. Also, what is frequent SAR acquisitions? Minutes, hours?

Line 101: " InSAR deformation is the projection of the three-dimensional ice movement…" Here, I am also getting a little lost. You mean InSAR-derived deformation? You can derive this in several steps. I think you are suggesting that: "initial motion derived directly from the interferometric phase represents the projected motion…". Furthermore, be careful so you don't mix deformation and movement. These are different and confused many places in the manuscript.

Line 102: This is a long sentence. Please split up and explain.

Line 105: "From this perspective, we investigated InSAR derived deformation information observed from SAR satellites". Now I am confused. You say you will look for surface elevation, change as a result of ice thickness change? This is not deformation though, but rather a shift in surface elevation or displacement. Deformation is something that leads to changes in the structure.

Line 109: "generated using InSAR techniques" What is this? Either take out or specify.

Line 153: Why do you bring up the harbor? And where is it?

Line 176: What 90 days? Did you mention this earlier? Please remind me. And why exactly 90 days and what time of the year? Are you looking at one year or several?

Line 191: Please include enough info for someone to replicate your work. How much did you average?

Line 217: It looks like you are using different tracks for the comparison. So potentially different types of ice roughness. Why don't you use the same tracks? Based on where you use overlapping tracks, ALOS actually shows a higher correlation than Radarsat. If you discuss this later, I would still encourage you to provide a little more information here and the reference to a later discussion so that the reader doesn't assume that you think Sentinel is superior. See broader comments.

281: What is high coherence?

284: What is good coherence?

322: The land position fluctuates? Please elaborate.

I stop the detailed read here as this manuscript has to be totally rewritten (see broader comments). Further figure/table suggestions follow:

Table 2: Please take out NA. It makes me have to look for the values. Would be a lot easier to see without the NA clutter. I suggest you just leave them blank.

Figure 1: Correct Denmark to Greenland. Please either cut Figure 2 as the information is described in a couple of words. Or even better, merge Figure 1 and 2. The overview panel in Figure 2 can then be enlarged somewhat and show ice classes. Please also annotate here with Queen Maud Gulf and other names discussed in the text. Please also use east and west instead of left/right when you refer to directions in the text.

Figure 9: You should show 9a and b with similar x-axis and pick either symbols or colors to differentiate the data.

Figure 10: This figure should be in the data section under a subsection on the ice profiler ice thickness.

Figure 12: The y-axis label is not consistent with the blue graph as it doesn't indicate displacement. The figure says this is displacement, but the caption that this is deformation. Which is it?